# The Impact of Dialect Diversity on Rent-Free Farmland Transfers: Evidence from Chinese Rural Household Surveys

Shangpu Li [1], Ye Jiang [2], Biliang Luo [1,*] and Xiaodan Zheng [3]

[1] National School of Agricultural Institution and Development, South China Agricultural University, Guangzhou 510642, China; qspli@scau.edu.cn
[2] Department of Economics, Dickinson College, Carlisle, PA 17013, USA; jiangy@dickinson.edu
[3] College of Economics and Management, South China Agricultural University, Guangzhou 510642, China; 20222035001@stu.scau.edu.cn
[*] Correspondence: luobl@scau.edu.cn

**Abstract:** The rent-free farmland transfer that exists widely in China's rural areas is a topic worthy of attention. Particularly, the regional heterogeneity of its occurrences implies regional cultural heterogeneities. Using local dialects to proxy regional cultural features, this study applies econometric methods to examine the impacts of dialect diversity on rent-free farmland transfers. It also considers possible mechanisms through a mediation analysis, based on a combined two-year rural household survey dataset from the Guangdong and Jiangxi Provinces in 2015 and 2016. Robust estimation results reveal that dialect diversity increases the probability of rent-free farmland transfers at the household and village levels. According to the mediation analysis, dialect diversity influences villages' farmland abandonment, rural farmland market development, and the flexibility of farmland transfer contracts, which further affects rent-free farmland transfer. Rent-free farmland transfer depends on social trust and contracts' self-fulfilling advantages. Therefore, cultural and traditional factors should be taken into consideration, which would form beneficial interactions between the selections of rural farmland institutional arrangement and land rights policy implementations.

**Keywords:** rural household; farmland transfer; rent free; dialect diversity

## 1. Introduction

In recent years, the scale of China's farmland transfers has continued to expand. The total farmland transfer areas have increased from 56 million mu in 2006 to 557 million mu in 2021, and the farmland transfer rate has increased from 4.74% to 35.37%. At the same time, rent-free farmland transfers in China have become increasingly prominent [1]. Rent-free farmland transfers exist widely among relatives and friends within rural households. Taking factors, such as social connections, family bonds, and interconnections with acquaintances, into account, rural–urban migrant workers tend to transfer their owned farmland to their relatives or friends for free through oral agreements, which not only prevents farmland from being left uncultivated but also helps to maintain social relationships among acquaintances [2]. For example, the overall incidence rate of rent-free farmland transfer was 55.05% based on China's rural household survey data [3], which was particularly high and exceeded 70% in mountainous areas [4]. The rural land market in developing countries is relatively imperfect, and payment methods such as zero rent are more common [5]. The Chinese rural land market also faces the same situation, whereby price signals do not work to some extent.

Some early studies pointed out that rent-free farmland transfers reflect the pattern of difference in China's farmland transfers [6], raised concerns regarding farmers' demands for their land property rights in the process of industrialization and urbanization [3], and revealed issues regarding the high transaction costs in the rural farmland market [1]. The development of the rural farmland market needs close cooperation between formal institutional arrangements and informal systematic instructions, which require further investigation regarding the

possible mechanisms, resulting in rent-free farmland transfer. However, so far, little research has been conducted regarding the regional heterogeneous occurrences of rural households' rent-free farmland transfer or the factors affecting decision making.

Furthermore, according to two consecutive annual rural household surveys of the Guangdong and Jiangxi Provinces in 2015 and 2016, the incidence rate of the rent-free farmland transfer was heterogeneous across the regions and related to the regional distributions of people who speak different dialects. Each area that speaks a dialect is not only a geographical region but also a cultural entity. Discrete rural villages are connected through a common language and integrated into a living community [7]. There are extensive studies on Chinese land transfer, only some of which have paid special attention to the regional culture and informal institutions involved, which will have an impact on the farmland market [8–10] and rural settlements [11].

Cultural evolution follows its historical path, along which its meaning is continuously enriched. At the same time, heterogeneities in cultural progress start appearing and gradually intensify across various geographical regions. As an important part of local culture, regional dialects experience long-term evolution and exhibit geographical diversities. Dialect diversity can be used as a key variable to represent local culture, influencing individual or social economic behaviors [12–14]. Without a doubt, higher dialect diversity increases transaction costs and hinders cooperation and regional economic development. Therefore, exploring the effects of dialect diversity on rent-free farmland transfers is of great significance in unraveling the underlying mechanisms of the rural land market and its cultural basis.

This study follows a main strategy of using dialects to proxy local cultural patterns to discuss the relationship between regional heterogeneous cultural features and the rent-free farmland transfers as well as any possible explanatory mechanisms. Robust evidence reveals that local dialect diversity can significantly promote rent-free farmland transfers. Finally, mediation analysis regarding the possible explanatory mechanisms was conducted using the abandoned farmland rate, the degree of farmland market development, and the rate of flexible rural land transfer contract use as mediating variables. This study mainly contributes to the previous literature by empirically studying the potential impacts of cultural factors, such as dialect diversity, on rent-free farmland transfers. And it applies modern econometric methods, including the Heckman selection model and the instrumental approach, to solve potential sample selection issues, including the endogeneity concern of local dialect diversity and omitted variable bias in traditional OLS estimations. Furthermore, this study provides micro-level evidence regarding rural households' farmland transfers as well as possible determining factors, which is in contrast to the perspective from previous regional or macro-level studies.

The rest of this paper is organized as follows: Section 2 introduces the related factual background, the theoretical background, and hypotheses; Section 3 describes the data and methods used; Section 4 displays the main estimation results; Section 5 further investigates the possible mechanisms with a mediation analysis and robustness test; and Section 6 presents the discussions.

## 2. Factual Background and Theoretical Framework

### 2.1. Language and Regional Dialect

Language is formed and evolved spontaneously during the process of human daily interactions. Language reflects the general pattern of local people's thinking in a certain region or ethnic group and serves as the foundation of social institutions [15]. Customs, conventions, norms, and laws of human society all rely on language to fully exert their functions. Representative of a country's history and culture, language can also affect people's way of thinking and behaviors. Language affects people's choices in social activities through how they express culture [16], which gradually forms solidified behavior characteristics [17].

Chinese people usually have a strong local affection as they are influenced deeply by traditional Confucianism and local culture [18]. As a part of local culture, regional dialect is a specific variant of language, distinguished by regional differences, and speaking the same dialect is taken as evidence of similarities shared by people from the same local areas.

Thus, from the perspective of regional identity, dialect is an important symbol of regional qualities shared only by local regional residents.

Dialects are typically the foundation of community identity [19], particularly in agricultural communities where stable local villages are formed. China had a higher than 90% dialect-speaking population among farmers before 1949 [20]. Because the Chinese government began promoting the national standard Mandarin in 1982, forty-five percent of rural residents can now speak the language [21]. Nonetheless, dialects are still present and can be easily identified by even the smallest variations in their accent.

Chinese belongs to the Sino-Tibetan language family, including seven major dialects, such as Cantonese, Min, Hakka, Gan, Wu, Xiang, and Mandarin. Guangdong Province is a 'treasure trove of dialects', with Cantonese, Minnan, and Hakka dialects. A wide range of dialects are also used in Jiangxi Province, including Mandarin, Gan, Hakka, Minnan, Wu, and Jianghuai Mandarin [22]. The proportion of residents using dialects is relatively higher, as the national dialect usage situation report shows 96.58% in Guangdong and 96.15% in Jiangxi when they were young and first learned their hometown dialect [21]. Therefore, this paper focuses on Guangdong Province and Jiangxi Province, where, in comparison to the national scenario, the percentage of residents who speak dialects is relatively large.

*2.2. Dialect Evolution and Its Endogenous Discussion*

Dialect evolution is very slow, which has a geographical and historical path. First, geographical information such as the locations and shapes of mountains and rivers has significant impacts on the evolution of dialects in specific regions [23,24], and many mountains or rivers were an important historic basis of administrative boundaries in China [25]. It is natural that people living in the same administrative region have more frequent connections and share more similarities in living habits, which can make local culture, customs, and dialect gradually consolidate and develop their own regional features [26,27]. In addition, the locations and shapes of mountains and rivers can also affect the level of local transportation convenience in a specific region, which have influences on the integration of languages [28,29]. Mountains often block inter-region communications due to inadequate traffic access, which was especially in ancient times. People can speak different dialects or languages in places separated by one rugged mountain. For example, Mountain Luofu, the geographic boundary between City Huizhou and Guangzhou, also separates regions speaking Cantonese and regions speaking Hakka [30].

Language communications of different species are also ecologically adaptive and impacted by climate factors [31,32], such as ambient humidity [33]. According to a study of over 4000 language varieties in the world, languages adopted in more humid areas have a reduced reliance on vowels compared to consonants [34]. In view of the connections between ambient humidity and languages, this study also applies them as instrumental variables.

Additionally, historical factors that affect the process of dialect evolution should have a limited relationship with current farmland transfers in our rural household survey. Comparing local dialects and Mandarin, we found that each local dialect followed a historical path during its evolution process [35]. The historical levels of each dialect are not only related to the immigration history but also closely connected to the local land resource allocations on the basis of the geographical distribution of dialects. From a long-term perspective, the evolution histories of some major Chinese local dialects, including Cantonese, Min, and Hakka, do support the relationship between immigration, dialect, and local land resource distributions [36–38]. Given this, this study also applies historical levels of each major dialect category as the instrumental variable.

*2.3. Rent-Free Farmland Transfer*

China has always attached great importance to its agricultural production throughout its history due to its large population and relatively limited cultivated land resources [12]. In the past two decades, the Chinese government has taken a series of measures aimed at expanding the scale of agricultural land management, including passing the Rural Land

Contract Law and issuing regulations on the operation of rural land management rights from top-level Chinese government offices [1,39][1].

The scale of China's farmland transfer has continued to expand. As reported by the Ministry of Agriculture and Rural Affairs[2], the total areas of farmland transfer increased from 56 million mu in 2006 to 557 million mu in 2021, and the corresponding farmland transfer rate increased from 4.74% to 35.37%. At the same time, the phenomenon of rent-free farmland transfer is prevalent in rural China. Based on China's rural household survey data collected from fixed rural observation sites between 2003 and 2013, Wang et al. [40] showed that the overall incidence rate of rent-free farmland transfer was 55.05% among all types of farmland transfers in rural China. Particularly, the percentage of rent-free farmland transfer in mountainous areas such as Chongqing City went even higher and could exceed 70% [41]. Qian and Ji [42] found that about 30% of the farmland in Jiangsu, Hubei, Heilongjiang, and Guangxi Provinces was transferred without any rent from 2006 to 2013. Although informal land transfer like rent-free farmland transfer indicates the imperfect development of China's land rental market, the current prominent scale of land transfer without rent in rural China [43,44] is nonnegligible, and dialectal diversity is worth further investigations regarding the possible explanatory factors [45,46].

Our research team conducted two annual rural household surveys of Guangdong and Jiangxi Provinces in 2015 and 2016 and found that the proportion of rent-free farmland transfer was 30.22%, exhibiting strong regional heterogeneities, which was also associated with the geographical distribution of dialect spoken in local areas. As shown in Figure 1, occurrences of rent-free farmland transfers are different across various geographical dialect regions in both Guangdong and Jiangxi Provinces. Generally, the proportion of rent-free farmland transfers is much higher within regions where people mainly speak Cantonese or Jiangxi local dialect (Gan). Additionally, rent-free farmland transfers perform differently across small dialect fragment areas[3]. For example, although the Siyi dialect fragment and Guangfu dialect fragment both belong to the Cantonese region, the percentage of rent-free farmland transfer in Siyi is lower than that in Guangfu.

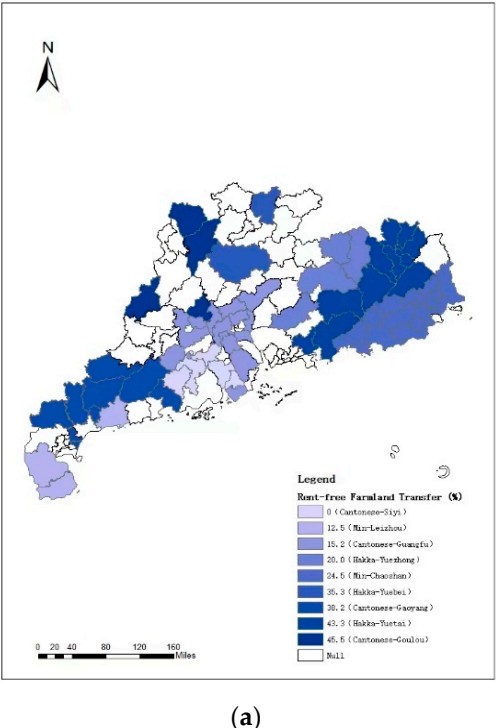
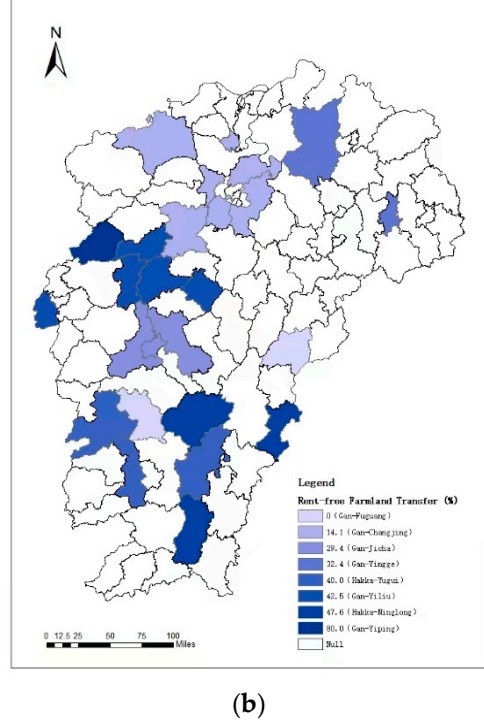

(**a**)                                                    (**b**)

**Figure 1.** Incidence rate of rent-free farmland transfers in dialect area. (**a**) Description of incidence rate of rent-free farmland transfers in dialect area in Guangdong province; (**b**) description of incidence rate of rent-free farmland transfers in dialect area in Jiangxi province.

### 2.4. The Relationship between Dialect Diversity and Rent-Free Farmland Transfer

Using dialect diversity as a proxy for the heterogeneous distribution of local dialects, this study mainly connects dialect diversity with rent-free farmland transfer through the social identity model since regional dialect conveys part of the signals related to regional identity. Akerlof and Kranton [47] introduced social identity into the utility analysis framework in standard neoclassical economics. According to them, individual utility maximization depends on individual behaviors, cohort companions' effects, and self-recognition, which associates social identity recognition and unique social norms within a cohort. Amartya [48] further pointed out that the individual identity recognition standard of other cohort companions had become a "loyalty filter", which can strongly affect personal behaviors as well as social interactions. On the one hand, the shared cohort identity contributes to building a better community within the specific cohort or region. On the other hand, unpleasant factors such as xenophobia against people from different identity groups do exist and can make changes to individual decisions.

Differences in dialects, ethnicities, and broad cultural backgrounds can lead to a lack of identity recognition [49] as well as social trust across people from various regions [50]. People who speak the same dialect can naturally find shared similarities originating from the same geographical region, which helps them build work relationships more efficiently. The "acquaintance society" is essentially a cultural phenomenon. On the contrary, people who speak different dialects or languages within the same regional market increase the probability of unpleasant communications as well as transaction costs [51], which hinders economic development, slows technology spillovers, and inhibits inter-regional movement of productive factors and the urbanization process at the macro level [52–55]. Challenges in local languages also prevent lots of non-native workers from entering better-remunerated industries and inter-provincial migrations [11,56].

### 2.5. Hypothesis

As mentioned above, dialect not only serves as a communication medium but also signals the trustworthiness of both transaction parties, during the process of local farmland transfer. The increasing regional dialect diversity naturally raises the bar for communication and inevitably worsens conceptual distance within the regional cohort. But, at the same time, the regional weakened social trust environment makes the specific trust originated from speaking the same dialect more prominent. As a type of "loyalty filter", speaking the same dialect helps to recognize the shared similarities among regional residents. Living in a region with a more diverse language environment, the shared similarities recognized through local dialects are more likely to be cherished, and a higher regional dialect diversity can induce more free giveaways that usually exist among acquaintances, including rent-free farmland transfer. As shown in Figure 1, various dialect fragment areas also exhibit regional disparities in rent-free farmland transfers, which indicates the possible connections between dialect diversity and rent-free farmland transfers.

**H1.** *Dialect diversity has a positive impact on rent-free farmland transfer.*

The interlinked contract exists widely in the agriculture economy [57]. Rural land is used to strengthen social relationships among acquaintances in the imperfect division of the labor market, while it sometimes serves as the investment of social capital and safety networks. Farmland transfer contracts have multidimensional factors, including rent, tenancy duration, lease status, etc. The social interaction literature emphasizes that individual interactions often arise in the same space, such as living in the same community, having the same identity, speaking the same dialect, etc. Agricultural land transfer transactions occurring between farmers in the same village imply infinite repeated game expectations, which effectively constrain opportunistic behaviors. The culture would affect the interlinked contract, such as decreasing farmland abandonment, cultivated land protection, and farmland transfer contracts, which bring the compensation rent free.

**H2.** *Dialect diversity has a positive impact on decreasing farmland abandonment and increasing contract flexibility for the compensation of rent-free farmland transfer.*

This indicates the fact that the development of rural farmland marketization is an alternative to traditional rural farmland transfers. With the gradual development of the rural land market, the ratio of rent-free farmland transfers was 15.5% in 2018 [53]. It is generally believed that higher dialect diversity increases transaction costs and hinders cooperation. Based on previous research, the factors affecting farmland transfer also influence rent-free farmland transfer. The agricultural land transfer market has obvious regional characteristics, within or between adjacent villages, and has a similar culture. Dialect diversity decreases trust or cooperation and hinders rural land marketization. When there are many potential trading partners in an active rural land market, farmers could find suitable leasing partners and achieve various interests, such as protecting farmland, instead of rent free. Thus, promoting rural farmland marketization decreases the incidence rate of rent-free farmland transfers.

**H3.** *Dialect diversity has a negative impact on farmland marketization, which is an alternative to rent-free farmland transfer.*

## 3. Materials and Methods

### 3.1. Data Source

The rural household data used in this paper come from two consecutive rural household sample surveys conducted within Guangdong and Jiangxi Province in 2015 and 2016. Surveys were organized by the National School of Agricultural Institution and Development (NSAID) of South China Agricultural University (SCAU) and funded by one major project, "Research on the Cultivation and Reform of Rural Land and Related Factor Markets", approved by the National Natural Science Foundation of China. Firstly, it used a random sampling method to select 24 counties in Guangdong province and 31 counties in Jiangxi province. Secondly, it adopted a random sampling method to select 15 townships from each county based on their level of economic development. Thirdly, a village was randomly selected in each township, and 10 farmers were randomly selected based on their income level. Finally, we collected a total of 4172 questionnaires, and 99.6% of them, 4156 questionnaires, were effective.

In terms of the dialect information, it mainly comes from the "Chinese Dialect Dictionary" and the "Chinese Language Atlas (Chinese Dialect Volume)". According to those two books, all Chinese dialects can be divided into 9 major categories. Those 9 major categories can then be subdivided into 80 sub-class dialects. The dialect fragment areas in the analytical sample are identified based on those 80 sub-class dialects spoken in each local area.

### 3.2. Key Variables

This study aims to estimate the impacts of dialect diversity on the occurrence of rent-free farmland transfer. We build our key explanatory variable, the dialect diversity index, through two distinct approaches. One approach is to measure the absolute dialect diversity (Dialect Diversity I) according to the number of dialect fragments in prefecture-level cities. The other approach is to measure the relative dialect diversity, which addresses overestimated issues when ignoring some factors such as area population. Thus, this study constructs the dialect diversity index (Dialect Diversity II) taking the population of each dialect fragment as weight,

$$\text{lang\_w}_i = 1 - \sum_{j=1}^{N} S_{ij}^2 \tag{1}$$

where $S_{ij}$ denotes the share of the population using dialect j in the city i. $S_{ij}^2$ is the squared of $S_{ij}$, and N is the number of dialect fragments in the city i. The range of $\text{lang\_w}_i$ is between 0 and 1, and a larger $\text{lang\_w}_i$ means a more diverse dialect environment in the city i.

Rent-free farmland transfer is not randomly distributed. It is the subsamples of transfer farmland, which encounter a statistical phenomenon of "tail breakage". It applies the Heckman selection model to solve the sample selection issues. The Heckman selection model divides rent-free transfer using a two-stage approach. The first stage is whether farmland transfers out occur in households or the village. The second stage is the occurrence of rent-free farmland transfer, by calculating the proportion of rent-free farmland transfer to farmland transfers out at the household or village level.

This study observes the rent-free farmland transfer at the household farmland level and village level. First, if farmers chose to rent out their farmland for free, then the binary choice variable "Rent-free Farmland Transfer" equals 1 for the interviewed household. Otherwise, it equals 0. Second, this study calculates the proportion of rural households that ever chose rent-free farmland transfers in all rural households in each village. This village rent-free farmland transfer percentage can reveal farmers' average willingness to rent their farmland out for free in each local village. Those two rent-free farmland transfer variables are the two main outcome variables used in later empirical analysis. This study also controls for some other covariate variables in regressions of rent-free farmland transfers, including some characteristic variables of farmland and rural villages. Details of the selected variables are displayed in Table 1[4].

**Table 1.** Variables and descriptive statistics.

| Variable | Definition | Measurements | N | Mean | Std. Dev. |
|---|---|---|---|---|---|
| Rent-free Farmland Transfer | 0/1; equals 1 if farmers chose to rent out their farmland for free at the second stage, otherwise, it equals 0 | 0/1 | 1077 | 0.301 | 0.459 |
| Farmland Transfer-out (%) | 0/1; equals 1 if farmers chose to rent their farmland at the first stage, otherwise, it equals 0. | 0/1 | 4177 | 0.256 | 0.436 |
| Village Rent-free Farmland Transfer (%) | # of households ever with rent-out-free farmland transfers/# of total household farmland transfers out in a village | % | 3138 | 0.347 | 0.363 |
| Village Farmland Transfer-out (%) | 0/1; equals 1 if Village rents out farmland at the first stage, otherwise, it equals 0 | 0/1 | 4177 | 0.751 | 0.432 |
| Dialect Diversity I | Number of dialect fragments in each city | Number | 4177 | 2.068 | 1.042 |
| Dialect Diversity II | The dialect Diversity index for each city calculated through Equation (1) | index | 4177 | 0.319 | 0.288 |
| Urban Land Price | The logarithm of land price for nonagricultural construction land | - | 4172 | 7.157 | 0.700 |
| Neighborhood Preferences | % of farmland transferred to neighbors or relatives in each village | % | 4172 | 0.098 | 0.146 |
| Farmland Protection Attitude (%) | % of households opposing changing the use of farmland in each village | % | 4177 | 0.633 | 0.295 |
| Regional farmland Protection Attitude (%) | % of households opposing changing the use of farmland in each town | % | 4172 | 0.473 | 0.437 |
| Land Fertility | 0/1; equals 1 if the land is rated as fair or above, otherwise, it equals 0 | 0/1 | 4172 | 0.363 | 0.481 |
| Land Irrigation | 0/1; equals 1 if the land's irrigation is rated as fair or above, otherwise, it equals 0 | 0/1 | 4172 | 0.416 | 0.493 |
| Land Contiguity | 0/1; equals 1 if the distance is relatively small, otherwise, it equals 0 | 0/1 | 4172 | 0.400 | 0.490 |
| Land Scale | Village average contracted land areas (mu) held by each household | mu | 4172 | 1.005 | 0.806 |
| Village Transportation | 0/1; equals 1 if the village transportation condition is rated as fair or above, otherwise, it equals 0 | 0/1 | 4172 | 0.536 | 0.401 |
| County Economic Development | Average rating about county economic development by all interviewed households in each village | - | 4172 | 2.890 | 0.827 |
| Farmer Percentage (%) | Average % of farmers in household labor (village-level) | % | 4172 | 0.356 | 0.181 |
| Farmland Adjustment (%) | % of adjusted farmland in each village | % | 4172 | 0.235 | 0.347 |
| Entitled Farmland | % of entitled farmland in each village | % | 4172 | 0.616 | 0.395 |
| Subsidy Acknowledgement | % of households knowing grain subsidy in each village | % | 4177 | 0.233 | 0.329 |

*3.3. Model*

3.3.1. Sample Selection Issue

The rent-free farmland transfer can only be observed when rural households have transferred their farmland out. By the time our rural household survey started, some rural households did not have farmland, or they did not transfer any of their farmland out. Thus, their observations regarding the rent-free farmland transfer are missing, which

may bring sample selection issues if we simply drop them from our analytical sample. Similar issues still exist when we study the relationship between dialect diversity and the rent-free farmland transfer at the village level, since some villages did not have any farmland transferred out.

To address the possible sample selection issue due to some households or villages without any farmland transfers, this study particularly fits the baseline maximum-likelihood probit model with sample selection and divides the rent-free farmland transfer process into two stages, selection to transfer farmland, and rent type decision. Other than those controlled covariates in the second-stage regression of rent-free farmland transfers, this study also considers the village-level incidence rate of farmland transfers to predict rural households' selections regarding farmland transfers. Similarly, to deal with the possible sample selection issue at the village level, this study applies the Heckman selection model to avoid potential bias to the traditional OLS estimations.

### 3.3.2. Baseline Regression

As the dependent variable rent out for free at the household level is a binary variable, this study used the Heckprobit model. The Heckprobit model can solve the selective bias problem through two-step regression. Thus, we estimate the farmland transfer out using an equation to calculate the inverse Mills ratio (athrho), and then add it as a control variable into the rent-free equation for re-estimation. This study first explores the relationship between regional dialect diversity and rent-free farmland transfers at the household level. The baseline estimation model is listed in Equation (2):

$$\text{Probit}(\text{Ltr}_{ic}) = \phi + \rho \times \text{Dia}_c + Z' \times \delta + \xi_{ic} \tag{2}$$

$$\text{Probit}(Y_{ic}) = \alpha + \beta \times \text{Dia}_c + X' \times \gamma + V \times \text{athrho} + \varepsilon_{ic} \tag{3}$$

This study mainly applies the probit model to estimate the impacts of dialect diversity on rural households' probability of farmland transfers out in Equation (2), where $\text{Ltr}_{ic}$ is the dependent variable and it is a binary choice variable. It equals 1 if a rural household $i$ transfers out the farmland living in the city $c$; otherwise, it equals 0. $\text{Dia}_c$ is the main explanatory variable, dialect diversity index of the city $c$, where household $i$, currently is. $Z'$ is a vector of covariates that describes specific characteristics of household, farmland, and village, as listed in Table 1. $\xi_{ic}$ is the error term.

In Equation (3), $Y_{ic}$ is the dependent variable for rent-free farmland transfers, and it is a binary choice variable. It equals 1 if the rural household $i$ living in the city $c$, ever transferred farmland out for free; otherwise, it equals 0. $\text{Dia}_c$ is the main explanatory variable, dialect diversity index of the city $c$, where household $i$, currently lives. X is a vector of covariates that describes specific characteristics of household farmland and village, as listed in Table 1. $\varepsilon_{ic}$ is the error term.

We also apply the Heckman selection model to study the rent-free farmland transfer at the village level. Then, the dependent variable $Y_{ic}$ represents the proportion of rent-free farmland transfers in all types of farmland transfers at a specific village $i$. STATA 16 software is used throughout our analysis.

### 3.3.3. Instrumental Variable Approach

In Equation (1), though the language variable, dialect changes slowly with time and is often taken as an exogenous variable, so its current distribution in a certain region may still correlate with some unobserved factor that may also affect local rural households' economic behaviors, including the rent-free farmland transfers. Omitting those unobserved variables related to local culture or inter-region migration can bring bias to the estimation results. This study mainly applies an instrumental variable approach to address the endogeneity concern of the dialect diversity index in the regressions of rent-free farmland transfers.

In this study, the degree of land relief reflecting regional topographic features is used as an instrumental variable for dialect diversity in each city due to its close relationships

with local dialect formation and its exogenous properties. Other than this geographical variable, the number of secondary tributary or above in each surveyed county recorded through the ArcGIS is also taken as an instrumental variable of dialect diversity in later two-stage least square estimations. This study also applies annual local average sunshine hours and precipitation in our survey areas from 2000 to 2019 as instrumental variables of dialect diversity. STATA 16 software is used for analysis. Finally, historical instrumental variables of dialect diversity are language distance, local opera, and war confrontations, and corresponding estimation results are discussed in later sections.

## 4. Results

### 4.1. Baseline Estimation Results

This study applies the Heckman selection model for benchmark regression. The Athro index is statistically significant in Table 2, which is appropriate for choosing the Heckprobit model for estimation.

This study first estimates the impacts of dialect diversity on rent-free farmland transfers based on the probit model. The corresponding estimation results are listed in Table 2. Columns (1) and (2) report the estimation results using rural households' probabilities of choosing rent-free farmland transfers, and columns (3) and (4) are based on model specifications using villages' proportion of rent-free farmland transfers as the dependent variable.

As shown in Table 2, dialect diversity can significantly increase rural households' probabilities of choosing rent-free farmland transfers as well as the proportion of rural households that ever chose rent-free farmland transfers in all rural households within each village. The coefficient estimates of Dialect Diversity I and Dialect Diversity II in columns (1) through (4) are all positively significant, which yields consistent evidence that dialect diversity plays an important role in affecting rural households' farmland transfer decisions. The rent-free farmland transfer is essentially an interconnected contract arrangement within local rural communities. Dialect is not only a language medium for communication and coordination but also conveys signals of trustworthiness between both parties during a transaction. A higher dialect diversity increases the average local transaction cost, but, at the same time, it strengthens the recognition of a common identity, represented by speaking the same dialect. Thus, in a more dialect-diverse region, rural households prefer to communicate and cooperate with their acquaintances, including the rent-free farmland transfers or other free giveaways.

In Table 2, estimates of covariates are also as expected. First, a higher urban land price in a city decreases the probability of rent-free farmland transfers significantly, which indicates the spillover effects of a higher urban land price on rural households' farmland transfer decisions. Second, coefficient estimates of Neighborhood Preferences are positively significant, which reflects rural households preferring to transfer farmland to neighbors or close relatives. Third, rural households' attitude regarding protecting their farmland is also important in promoting incidences of rent-free farmland transfers since the variable Farmland Protection Attitude is also positively significant. Fourth, the farmland conditions in each village also have significant impacts on the rent-free farmland transfers since variables, including Land Scale, Land Fertility, and Land Irrigation, are determining factors of the farmland rent. Additionally, the economic development level in local counties, rural households' acknowledgment of grain subsidies, and the adjustment of farmland in each village also have nonnegligible impacts on the rent-free farmland transfers.

**Table 2.** Impacts of dialect diversity on rent-free farmland transfers based on Heckprobit and Heckman model.

| | Household-Level | | Village-Level | |
|---|---|---|---|---|
| | **(1)** | **(2)** | **(3)** | **(4)** |
| Dialect Diversity I | 0.1030 ** | | 0.0449 *** | |
| | (0.0433) | | (0.0069) | |
| Dialect Diversity II | | 0.3039 ** | | 0.1565 *** |
| | | (0.1503) | | (0.0261) |
| Urban Land Price | −0.1152 * | −0.1208 * | −0.0377 *** | −0.0403 *** |
| | (0.0669) | (0.0664) | (0.0111) | (0.0112) |
| Neighborhood Preferences | 1.9481 *** | 1.9496 *** | 0.5446 *** | 0.5776 *** |
| | (0.2011) | (0.2000) | (0.0573) | (0.0514) |
| Farmland Protection Attitude (%) | 0.2867 * | 0.3151 * | −0.0429 ** | −0.0356 * |
| | (0.1660) | (0.1625) | (0.0195) | (0.0197) |
| Land Fertility | −0.1133 | −0.1150 | −0.0645 *** | −0.0611 *** |
| | (0.1014) | (0.1000) | (0.0167) | (0.0169) |
| Land Irrigation | −0.0398 | −0.0376 | −0.0326 *** | −0.0332 ** |
| | (0.0963) | (0.0959) | (0.0164) | (0.0165) |
| Land Contiguity | 0.0068 | 0.0058 | −0.0390 *** | −0.0402 *** |
| | (0.0779) | (0.0774) | (0.0126) | (0.0126) |
| Land Scale | −0.2488 *** | −0.2414 *** | −0.0880 *** | −0.0778 *** |
| | (0.0699) | (0.0600) | (0.0117) | (0.0118) |
| Village Transportation | −0.1461 | −0.1360 | −0.0872 *** | −0.0678 *** |
| | (0.1053) | (0.1050) | (0.0184) | (0.0199) |
| County Economic Development | −0.1130 * | −0.1224 ** | −0.0678 *** | −0.0857 *** |
| | (0.0598) | (0.0587) | (0.0109) | (0.0126) |
| Farmer Percentage (%) | −0.3901 | −0.4122 | −0.12636 *** | 0.0886 |
| | (0.3305) | (0.3274) | (0.0504) | (0.0540) |
| Farmland Adjustment (%) | −0.2913 ** | −0.2776 ** | 0.0883 *** | −0.2187 *** |
| | (0.1215) | (0.1202) | (0.0250) | (0.0199) |
| Entitled Farmland | 0.1095 | 0.1075 | −0.0595 | 0.1003 *** |
| | (0.1179) | (0.1174) | (0.0369) | (0.0257) |
| Subsidy Acknowledgement | −0.1602 | −0.1817 | −0.0595 | −0.0986 ** |
| | (0.1738) | (0.1725) | (0.0369) | (0.0394) |
| Athrho | 1.0555 *** | 1.0740 *** | 0.6131 | 0.3994 *** |
| | (0.2088) | (0.2088) | (0.0440) | (0.1038) |
| Constant | −0.2306 | −0.0734 | 0.8090 *** | 0.8756 *** |
| | (0.6255) | (0.6116) | (0.1016) | (0.1009) |
| Wald Chi-square | 156.42 | 155.76 | 541.29 | 578.65 |
| *p*-value | (0.0000) | (0.0000) | (0.0000) | (0.0000) |
| Log pseudolikelihood | −2253.915 | −2254.829 | −1787.58 | −2263.628 |
| Wald Test | 25.56 | 26.45 | 1.94 | 14.08 |
| *p*-value | (0.0000) | (0.0000) | 0.1635 | 0.0001 |
| N | 4172 | 4172 | 4172 | 4172 |

Notes: Huber-White robust standard errors, in parentheses. *** $p < 0.01$, ** $p < 0.05$, * $p < 0.1$.

*4.2. Two-Stage Least Square (2SLS) Estimation Results*

Table 3 displays the 2SLS estimation results using the degree of land relief of each city as the instrumental variable in regressions of rent-free farmland transfers. Similar to Table 2, estimations in columns (1) and (2) are based on household level, and columns (3) and (4) are at the village level. The lower panel of Table 3 reports the estimation results of the first stage, which examines the correlation between the instrumental variable and the endogenous variable, dialect diversity I or II. As shown in Table 3, coefficient estimates of the instrumental variable, the degree of land relief, are all statistically significant in the first-stage regressions, which satisfies the correlation assumption of instrumental variables. The F-statistics associated with the test of the hypothesis show that the coefficients on the excluded instruments are jointly equal to zero and are all above the common criteria, which prove that the degree of land relief is a powerful instrument for dialect diversity. The

coefficient estimates of the dialect diversity in the second-stage regressions are all positively significant, which provides consistent evidence with the baseline estimations.

**Table 3.** Impacts of dialect diversity on rent-free farmland transfers based on two-stage least squares method (IV: The Degree of Land Relief).

| | Household-Level | | Village-Level | |
|---|---|---|---|---|
| **2nd Stage** | **(1)** | **(2)** | **(3)** | **(4)** |
| Dialect Diversity I | 0.0779 ** | | 0.0962 *** | |
| | (0.0325) | | (0.0133) | |
| Dialect Diversity II | | 0.2699 *** | | 0.3321 *** |
| | | (0.1129) | | (0.0484) |
| Constant | 0.4374 | 0.5014 | 0.5164 *** | 0.6466 *** |
| | (0.2516) | (0.2361) | (0.1086) | (0.1009) |
| Control Variables | Yes | Yes | Yes | Yes |
| Wald Chi-square | 133.69 | 132.82 | 855.48 | 864.92 |
| $p$-value | (0.0000) | (0.0000) | (0.0000) | (0.0000) |
| Log pseudolikelihood | −1914.0993 | −527.5866 | −5583.0451 | −1466.8687 |
| Wald Test | 5.75 | 5.71 | 52.49 | 47.02 |
| $p$-value | (0.0165) | (0.0168) | 0.0030 | 0.0039 |
| Weak Instrument Test | 5.72 | 5.71 | 52.15 | 47.03 |
| $p$-value | (0.0167) | (0.0169) | (0.0000) | (0.0000) |
| DHW Test | —— | —— | 14.4089 *** | 15.9135 *** |
| $p$-value | —— | —— | (0.0000) | (0.0000) |
| N | 1072 | 1072 | 3133 | 3133 |
| **1st Stage** | | | | |
| Degree of Land Relief | 3.5371 *** | 1.0217 *** | 3.6371 *** | 1.0029 *** |
| | (0.1932) | (0.0556) | (0.1116) | (0.0314) |
| Constant | 1.3959 *** | −0.1667 | 1.3483 *** | 0.1049 *** |
| | (0.3782) | (0.1088) | (0.2269) | (0.0635) |
| Control Variables | Yes | Yes | Yes | Yes |
| $R^2$ | 0.5267 | 0.4973 | 0.5069 | 0.4813 |
| F-statistic | 78.35 | 69.63 | 215.94 | 192.83 |
| $p$-value | (0.0000) | (0.0000) | (0.0000) | (0.0000) |
| N | 1072 | 1072 | 3133 | 3.133 |

Notes: Huber–White robust standard errors, in parentheses. *** $p < 0.01$, ** $p < 0.05$.

As mentioned earlier, this study also applies other instrumental variables, including annual average sunshine hours, precipitation in each city between 2000 and 2019, and the number of secondary or above tributaries in each surveyed county. Table 4 reports the 2SLS estimation results in the regressions of Dialect Diversity II[5]. Relatively, the estimation results using sunshine hours as the instrument are more robust, indicating that a higher local dialect diversity can induce more rent-free farmland transfers. In columns (3) through (6), though coefficient estimates of the dialect diversity are not uniformly statistically significant, estimation results using precipitation or the number of tributaries as the instrumental variables still provide consistent evidence, proving that dialect diversity has positive impacts on the rent-free farmland transfers.

Table 5 reports the 2SLS estimation results using the historical levels of each major dialect category as the instrumental variable. Consistent with the baseline estimation results, the local dialect diversity can significantly promote rent-free farmland transfers at the household level and village level, which also indicates that the "principle of preemption" in the original land resource allocations, as shown by dialect historical levels, reflects the historical connections of social identity recognitions and regional cultural patterns.

**Table 4.** Impacts of dialect diversity on rent-free farmland transfers based on two-stage least squares method (IV: Sunshine Hours, Precipitation, # of Tributaries).

| | Sunshine Hours | | Precipitation | | # of Tributaries | |
|---|---|---|---|---|---|---|
| **2nd Stage** | **Household (1)** | **Village (2)** | **Household (3)** | **Village (4)** | **Household (5)** | **Village (6)** |
| Dialect Diversity II | 0.5032 *** | 0.4394 *** | 2.8383 | 0.3774 | 0.1360 | 0.3343 *** |
| □ | (0.1494) | (0.0622) | (0.3.0761) | (0.2682) | (0.1489) | (0.0720) |
| Constant | 0.5777 | 0.4568 *** | −2.5666 | 0.5926 | 0.6615 | 0.6527 *** |
| □ | (0.3339) | (0.1104) | (3.691) | (0.330) | (0.2626) | (0.1204) |
| Control Variables | Yes | Yes | Yes | Yes | Yes | Yes |
| Wald Chi-square | 135.87 | 872.54 | 169.28 | 812.20 | 128.75 | 1227.56 |
| □ | (0.0000) | (0.0000) | (0.0000) | (0.0000) | (0.0000) | (0.0000) |
| Log Pseudolikelihood | −581.2864 | −1539.0271 | −750.0119 | −1914.017 | −724.13584 | −458.07104 |
| Wald Test/LM Test | 11.34 *** | 49.95 *** | 5.41 | 1.98 | 110.76 | 21.58 |
| *p*-value□ | (0.0008) | (0.0000) | (0.0200) | (0.1594) | (0.0000) | (0.0000) |
| Weak Instrument Test | 11.76 *** | 51.50 *** | 2.74 | 2.00 | 0.04 | 21.69 |
| *p*-value□ | (0.0006) | (0.0000) | (0.0981) | (0.1574) | (0.8481) | (0.0000) |
| DHW Test | —— | 28.53 | —— | 0.6410 | —— | 7.3015 |
| *p*-value | —— | (0.0000) | —— | (0.4234) | —— | (0.0069) |
| N | 1072 | 3133 | 1072 | 3133 | 1072 | 1072 |
| **1st Stage** | | | | | | |
| Instrument | −0.0013 *** | −0.0013 ** | −0.0001 | −0.00002 *** | 0.1628 *** | 0.1426 *** |
| □ | (0.0001) | (0.001) | (0.0000) | (0.0000) | (0.0125) | (0.0070) |
| Constant | 1.6565 *** | 3.0591 *** | 1.3093 *** | 1.5011 *** | 0.2506 *** | 1.2992 *** |
| □ | (0.2102) | (0.1003) | (0.1529) | (0.0866) | (0.0996) | (0.0583) |
| Control Variables | Yes | Yes | Yes | Yes | Yes | Yes |
| $R^2$ | 0.5015 | 0.4239 | 0.3371 | 0.3176 | 0.4285 | 0.3905 |
| F-statistic | 70.83 | 152.88 | 35.8 | 96.71 | 52.79 | 142.72 |
| *p*-value | (0.0000) | (0.0000) | (0.0000) | (0.0000) | (0.0000) | (0.0000) |
| N | 1072 | 3133 | 1072 | 3133 | 1072 | 3133 |

Notes: Huber–White robust standard errors, in parentheses. *** $p < 0.01$, ** $p < 0.05$.

**Table 5.** Impacts of dialect diversity on rent-free farmland transfers based on the two-stage least squares method (IV: Historical Levels of Each Major Dialect Category) (IV: Historical Levels of Each Major Dialect Category).

| | Household-Level | | Village-Level | |
|---|---|---|---|---|
| **2nd Stage** | **(1)** | **(2)** | **(3)** | **(4)** |
| Dialect Diversity I | 0.268 ** | | 0.158 *** | |
| | (0.110) | | (0.028) | |
| Dialect Diversity II | | 0.891 ** | | 0.734 *** |
| | | (0.385) | | (0.08) |
| Constant | −0.228 | 0.048 | 0.253 | 0.223 * |
| | (1.105) | (1.131) | (0.160) | (0.130) |
| Control Variables | Yes | Yes | Yes | Yes |
| Wald Chi-square | 117.69 *** | 119.33 *** | 655.15 *** | 648.41 *** |
| Log Pseudolikelihood | −2029.756 | −662.941 | 0.112 | 0.127 |
| Wald Test | 5.30 ** | 4.01 ** | 204.244 *** | 247.906 *** |
| *p*-value | (0.021) | (0.045) | (0.000) | (0.000) |
| Weak Instrument Test | 11.52 *** | 6.03 ** | 31.91 *** | 32.46 *** |
| *p*-value | (0.001) | (0.014) | (0.000) | (0.000) |
| DHW Test | - | - | 20.058 *** | 76.469 *** |
| *p*-value | | | (0.000) | (0.000) |
| N | 1072 | 1072 | 1072 | 1072 |
| **1st Stage** | | | | |
| Dialect Historical Level | 0.261 *** | 0.073 *** | 0.1555 *** | 0.0500 *** |
| | (0.015) | (0.004) | (0.011) | (0.003) |
| Constant | 0.744 *** | −0.0519 ** | 3.0728 *** | 0.6167 *** |
| | (0.084) | (0.024) | (0. 247) | (0.066) |
| Control Variables | Yes | Yes | Yes | Yes |
| $R^2$ | 0.5341 | 0.4873 | 0.3916 | 0.3897 |
| F-statistic | 80.72 *** | 66.92 *** | 143.36 *** | 132.67 *** |
| N | 4156 | 4156 | 4156 | 4156 |

Notes: Huber–White robust standard errors, in parentheses. *** $p < 0.01$, ** $p < 0.05$, * $p < 0.1$.

In the appendix, this study also selects some other instrumental variables, including the dialect distance [58], the number of local operas in each city [26,59], and the number of wars or armed confrontations in each city [60,61]. Corresponding 2SLS estimation results of the Dialect Diversity II are listed in Table A1 and are qualitatively consistent with previous estimation results.

## 5. Further Analysis

### 5.1. Possible Mechanisms

In this section, this study tries to further investigate possible mechanisms that may explain the positive impacts of dialect diversity on rent-free farmland transfers in rural China.

As China started its economic reform in 1978, an increasing number of rural laborers persistently moved to urban areas for better-remunerated jobs and higher living standards. The industrialization and urbanization process in recent years also promoted rural–urban migration. For those farmers who have decided to migrate to urban areas, possible handling strategies for their own contracted farmland include transfer with rent, transfer without rent, and being left uncultivated. The regional dialect diversity affects people's recognition level of original identity and trust in those from the same hometown, which further influences decisions about handling their farmland, including the rent-free farmland transfers.

As mentioned earlier, a higher regional dialect diversity increases difficulties in communication and cooperation, which can also affect local transactions and economic development. Considering the possible impacts of dialect diversity on factors that can affect rent-free farmland transfer, this study applies the mediation analysis to explore possible mechanisms that explain rural households' decisions regarding rent-free farmland transfers. Particularly, this study focuses on three possible mediator variables, including the rate of abandoned farmland, rural farmland market development level, and households' choices of flexible farmland transfer contracts.

Previous estimation results already proved a significant relationship between the dependent variable, the rent-free farmland transfers, and the key explanatory variable, with the dialect diversity based on Equation (1). Following Baron and Kenny's steps [62], we build our mediation models as follows:

$$M = \alpha + \beta \times \text{Dia} + X' \times \gamma + \varepsilon \tag{4}$$

$$Y = \alpha + \beta \times \text{Dia} + \delta \times M + X' \times \gamma + \varepsilon \tag{5}$$

can affect M, the mediator variable. Mediation makes sense only if the coefficient estimate of $\beta$ in Equation (4) is statistically significant. Later, to further investigate the mediation effects from M between the dialect diversity and the rent-free farmland transfer, we conduct estimations based on Equation (5), which include M and Dia on the right side simultaneously.

This study first examines the mediation effects of the village farmland abandonment rate between the dialect diversity and rent-free farmland transfer. The farmland abandonment rate in each village is a ratio between the abandoned farmland areas and the total contracted farmland areas owned by all rural households in each surveyed village. Table 6 reports the corresponding estimation results. In columns (1) and (2), we test the impacts of the dialect diversity on the village's abandoned farmland rate. Since speaking the same dialect works as a "loyalty filter" during the process of farmland transfer, the abandoned farmland rate of each village increases when rural households are less likely to find suitable partners to transfer their farmland to in a more diverse dialect-speaking environment.

The Sobel test results in Table 6 also indicate the statistical significance of the mediation effects of the village farmland abandoned rate between the dialect diversity and the rent-free farmland transfer. In columns (3) through (6), the coefficient estimates of the dialect diversity are still positively significant but smaller in magnitude than those listed in Table 2. At the same time, positively significant estimates of the village farmland abandoned rate reveal the fact that rural households tend to choose the rent-free farmland transfer, and the

proportion of rent-free farmland transfers in total farmland transfers rises as the village farmland abandoned rate increases. According to the mediation analysis of the village farmland abandoned rate, the rent-free farmland transfer also implies rural households' willingness to protect their farmland, since farmers who receive the farmland for free usually take active measures to maintain the land quality during their farming process and retribute the trust from original landowners.

**Table 6.** Mediation analysis of village farmland abandoned rate based on Heckprobit and Heckman model.

| | Village Abandoned Farmland Rate | | Households' Rent-Free Farmland Transfers | | Village Rent-Free Farmland Transfer (%) | |
|---|---|---|---|---|---|---|
| | **(1)** | **(2)** | **(3)** | **(4)** | **(5)** | **(6)** |
| Dialect Diversity I | 0.0078 * | | 0.0557 ** | | 0.0459 *** | |
| | (0.0040) | | (0.0148) | | (0.0064) | |
| Dialect Diversity II | | 0.0530 *** | | 0.1500 * | | 0.1461 *** |
| | | (0.0141) | | (0.0513) | | (0.0226) |
| Village Farmland Abandoned Rate | | | 0.3346 *** | 0.3311 *** | 0.4979 *** | 0.4005 *** |
| | | | (0.0674) | (0.0678) | (0.0336) | (0.0334) |
| Athrho | | | 1.0663 *** | 1.008 *** | 0.3176 *** | 0.3990 *** |
| | | | (0.2007) | (0.1881) | (0. 0990) | (0.128) |
| Constant | 0.3412 *** | 0.3188 *** | −1.6486 *** | −1.5606 | 0.3440 *** | 0.6077 *** |
| | (0.0492) | (0.0478) | (0.3711) | (0.3588) | (0.470) | (0.459) |
| Control Variables | Yes | Yes | Yes | Yes | Yes | Yes |
| Wald Chi-square | - | - | 178.00 *** | 175.97 *** | 812.09 *** | 761.89 *** |
| Log | - | - | −2249.215 | −2248.163 | −2085.039 | −2096.426 |
| Wald Test | - | - | 57.53 *** | 55.44 *** | 10.29 *** | 9.71 *** |
| *p*-value | - | - | (0.000) | (0.000) | (0.0013) | (0.002) |
| Sobel Z | - | - | 1.844 * | 2.624 *** | 2.263 ** | 2.266 *** |
| *p*-value | - | - | (0.0652) | (0.0088) | (0.0236) | (0.0235) |
| N | 4172 | 4172 | 4172 | 4172 | 3133 | 3133 |

Notes: Huber–White robust standard errors, in parentheses. *** $p < 0.01$, ** $p < 0.05$, * $p < 0.1$.

This study also examines the mediation effects of the local rural farmland market on rent-free farmland transfers. Table 7 reports the mediation effects of the rural farmland market development level between the dialect diversity and the rent-free farmland transfer. The rural farmland market development level is evaluated based on farmland transfer in and transfer out at the household level. We use the incidence rate of farmland transfers in a local town, a ratio between rural households who ever participated in a farmland transfer and total rural households, as a proxy for the rural farmland market development level.

As revealed in columns (1) and (2) of Table 7, the dialect diversity significantly lowers the development level of the local rural farmland market due to a lack of trust among rural households in a more diverse dialect-speaking community. Related Sobel tests also prove that the mediation effects of the rural farmland market development are statistically significant. Model specifications in columns (3) and (4) estimate the impacts of dialect diversity and rural farmland market development on rural households' decisions regarding rent-free farmland transfers. Columns (5) and (6) report related estimation results in the regressions of the percentage of village rent-free farmland transfers. The development level of the local rural farmland market can serve as a mediator, mainly because rural households can find more suitable transaction partners in an active rural farmland market. As shown in columns (5) and (6), a more developed rural farmland market lowers the incidence rate of rent-free farmland transfers in each village, which indicates that realizing rural farmland marketization is an alternative to traditional rural farmland transfers.

This study further investigates the mediation effects of flexible farmland transfer contracts between dialect diversity and rent-free farmland transfers. Model specifications in columns (1) and (2) use the ratio between the number of null or oral contracts and the

number of rural households in each surveyed village as the dependent variable, which reflects the rate of flexible farmland transfer contracts. As shown in columns (1) and (2), a higher dialect diversity promotes the rate of flexible farmland transfer contracts.

**Table 7.** Mediation analysis of rural farmland market development level based on Heckprobit and Heckman model.

| | Rural Farmland Market Development Level | | Households' Rent-Free Farmland Transfers | | Village Rent-Free Farmland Transfer (%) | |
|---|---|---|---|---|---|---|
| | **(1)** | **(2)** | **(3)** | **(4)** | **(5)** | **(6)** |
| Dialect Diversity I | −0.0130 *** (0.0038) | | 0.0455 ** (0.0158) | | 0.03707 *** (0.0608) | |
| Dialect Diversity II | | −0.0300 ** (0.013) | | 0.1172 ** (0.0552) | | 0.1297 *** (0.0237) |
| Rural Farmland Market Development Level | | | −0.4684 *** (0.1340) | −0.4753 *** (0.1343) | −0.4827 *** (0.0608) | −0.4831 *** (0.0608) |
| Athrho | | | 0.9132 *** (0.2328) | 0.9291 *** (0.233) | 0.2226 *** (0.004) | 0.2280 *** (0.0774) |
| Constant | −0.0441 (0.0463) | −0.0730 (0.0452) | 0.7145 *** (0.2097) | 0.8082 *** (0.2050) | 1.0988 *** (0.0903) | 1.1446 *** (0.0877) |
| Control Variables | Yes | Yes | Yes | Yes | Yes | Yes |
| Wald Chi-square | - | - | −156.16 *** | 155.15 *** | 864.66 *** | 847.25 *** |
| Log | - | - | −2253.362 | −2254.257 | −2132.465 | −2143.833 |
| Wald Test | - | - | 15.39 *** | 15.90 ** | 8.43 *** | 8.68 *** |
| *p*-value | - | - | (0.002) | (0.0000) | (0.0037) | (0.0032) |
| Sobel Z | - | - | 2.222 ** | 2.329 ** | −0.3911 | 0.4728 |
| *p*-value | - | - | (0.0263) | (0.0199) | (0.6957) | (0.6364) |
| N | 4172 | 4172 | 1072 | 1072 | 3133 | 3133 |

Notes: Huber–White robust standard errors, in parentheses. *** $p < 0.01$, ** $p < 0.05$.

The Sobel test results reveal the statistical significance of the mediation effects of flexible farmland transfer contracts. Columns (3) through (6) of Table 8 report the impacts of the dialect diversity and flexible farmland transfer contracts on the incidence of rent-free farmland transfers at the household level and village level, respectively. Flexible rural farmland transfer contracts, such as oral contracts, are often taken as latent rules during rural farmland transfers, which have self-fulfilling advantages. The rent-free farmland transfer is essentially an interconnected contract arrangement. Especially during China's current industrialization and urbanization process, rural farmland is generally expected to appreciate. Making more flexible contract arrangements is a form of "compensation" or a strategic choice for rent-free farmland transfers.

**Table 8.** Mediation analysis of flexible farmland transfer contracts based on Heckprobit and Heckman model.

| | Flexible Farmland Transfer Contracts | | Households' Rent-Free Farmland Transfers | | Village Rent-Free Farmland Transfer (%) | |
|---|---|---|---|---|---|---|
| | **(1)** | **(2)** | **(3)** | **(4)** | **(5)** | **(6)** |
| Dialect Diversity I | 0.032 *** (0.012) | | 0.091 * (0.049) | | 0.028 *** (0.007) | |
| Dialect Diversity II | | 0.081 * (0.042) | | 0.244 (0.165) | | 0.107 *** (0.025) |
| Flexible Farmland Transfer Contracts | | | 1.158 *** (0.137) | 1.379 *** (0.198) | 0.429 *** (0.024) | 0.448 *** (0.024) |
| Athrho | | | 0.713 *** (0.164) | 0.767 *** (0.185) | 0.353 *** (0.098) | 0.381 *** (0.099) |

**Table 8.** *Cont.*

| | Flexible Farmland Transfer Contracts | | Households' Rent-Free Farmland Transfers | | Village Rent-Free Farmland Transfer (%) | |
|---|---|---|---|---|---|---|
| | (1) | (2) | (3) | (4) | (5) | (6) |
| Constant | 0.533 *** | 0.648 *** | −0.607 | −0.636 | 0.723 *** | 0.738 *** |
| | (0.159) | (0.155) | (0.719) | (0.680) | (0.108) | (0.106) |
| Control Variables | Yes | Yes | Yes | Yes | Yes | Yes |
| Wald Chi-square | - | - | 198.45 *** | 180.24 *** | −1951.601 *** | 1656.85 *** |
| Log | - | - | −2253.738 | −2460.033 | 1476.47 | −1942.148 |
| Wald Test | - | - | 19.04 *** | 16.56 ** | 13.11 *** | 14.87 *** |
| *p*-value | - | - | (0.000) | (0.041) | (0.000) | (0.000) |
| Sobel Z | - | - | 0.014 *** | 0.030 *** | 0.010 *** | 0.020 ** |
| *p*-value | - | - | (0.002) | (0.050) | (0.000) | (0.026) |
| N | 1726 | 1726 | 1069 | 1069 | 3133 | 3133 |

Notes: Huber–White robust standard errors, in parentheses. *** $p < 0.01$, ** $p < 0.05$, * $p < 0.1$.

### 5.2. Robustness Checks

This study also conducts a series of tests to examine the robustness of previous estimation results. First, this study excludes rural households who ever participated in farmland transfer in and transfer out in the analytical sample. Instead, this study examines the impacts of dialect diversity on the rent-free farmland transfers within a subsample of rural households who only participated in farmland transfer out (Net Farmland Transfer). Second, this study also uses an alternative dependent variable, rural household's willingness to choose the rent-free farmland transfer, to replicate previous estimations and obtain consistent evidence. Last, this study clusters standard errors of coefficient estimates at the village level. As shown in Table 9, for the two main explanatory variables, Dialect Diversity I and Dialect Diversity II, the corresponding estimates are qualitatively consistent with previous OLS and 2SLS estimation results.

**Table 9.** Estimation results of robustness checks based on Heckprobit and Heckman model.

| | Net Farmland Transfers | | Willing of the Rent-Free Farmland Transfer | | Std. Clustered at Village Level | |
|---|---|---|---|---|---|---|
| | Household (1) | Village (2) | Household (3) | Village (4) | Household (5) | Village (6) |
| Dialect Diversity I | 0.1023 ** | 0.0241 *** | 0.2228 *** | 0.0658 *** | 0.1198 * | 0.0387 |
| | (0.0440) | (0.038) | (0.0558) | (0.0047) | (0.0718) | (0.0288) |
| Athrho | 1.200 *** | 0.2608 *** | 0.8106 *** | 0.1709 *** | 1.0104 ** | 0.3650 |
| | (0.2858) | (0.0344) | (0.1603) | (0.0633) | (0.2695) | (0.2902) |
| Constant | −0.4653 | 0.7632 *** | 0.0991 | 0.3224 *** | −0.0123 | 1.0123 |
| | (0.6337) | (0.0521) | (0.7514) | (0.0630) | (0.9385) | (0.3765) |
| Control Variables | Yes | Yes | Yes | Yes | Yes | Yes |
| Wald Chi-square | 178.06 | 106.95 | 125.35 | 1353.04 | 113.83 | 97.45 |
| *p*-value | (0.000) | (0.000) | (0.000) | (0.000) | (0.000) | (0.000) |
| Log pseudolikelihood | −2218.472 | −179.4405 | −2069.817 | −1104.977 | −2239.942 | −2150.86 |
| Wald Test | 17.65 | 57.42 | 25.56 | 7.30 | 14.06 | 1.58 |
| *p*-value | (0.0000) | (0.0000) | (0.000) | (0.0000) | (0.0002) | (0.2085) |
| N | 4172 | 4172 | 4172 | 4172 | 4160 | 4160 |
| Dialect Diversity II | 0.2952 ** | 0.10592 *** | 0.7520 *** | 0.2515 *** | 0.0771 *** | 0.149 |
| | (0.1528) | (0.0147) | (0.1941) | (0.0170) | (0.0321) | (0.103) |
| Athrho | 1.2255 *** | 0.2708 *** | 0.8584 *** | 0.2074 *** | 1.1322 *** | 0.2664 *** |
| | (0.2850) | (0.0349) | (0.1705) | (0.6221) | (0.2738) | (0.0955) |
| Constant | −0.2945 | 0.8209 *** | 0.3179 | 0.3771 *** | −0.1167 | 0.9009 |
| | (0.6210) | (0.0500) | (0.7291) | (0.0598) | (0.7400) | (0.1408) |
| Control Variables | Yes | Yes | Yes | Yes | Yes | Yes |
| Wald Chi-square | 177.62 | 93.93 | 123.91 | 1351.47 | 110.25 | 18.54 |
| *p*-value | (0.000) | (0.000) | (0.000) | (0.000) | (0.000) | (0.1833) |
| Log pseudolikelihood | −2219.569 | −200.6172 | −2070.722 | −1101.168 | −2239.344 | −169.417 |
| Wald Test | 18.49 | 60.30 | 25.36 | 11.12 | 17.09 | 7.7 |
| *p*-value | (0.000) | (0.000) | (0.000) | (0.0009) | (0.026) | (0.0053) |
| N | 4172 | 4172 | 4172 | 4172 | 4160 | 4160 |

Notes: Huber–White robust standard errors, in parentheses. *** $p < 0.01$, ** $p < 0.05$, * $p < 0.1$.

## 6. Discussions

### 6.1. Conclusions

Different from many countries in the world, China's rural farmland transfers have some unique features, including rent-free farmland transfer. From the perspective of cultural origin, this study aims to investigate the relationship between heterogeneous regional cultures and the incidence of local rent-free farmland transfers. Particularly, this study estimates the impacts of dialect diversity on rent-free farmland transfers based on two consecutive Chinese rural household sample surveys following a proxy strategy that heterogeneous distributions of dialects can represent various cultural characteristics.

The baseline estimation results reveal that regional local dialect diversity significantly promotes rent-free farmland transfer at the rural household level as well as at the village level. Later, 2SLS estimations apply exogenous geographical, climate, and historical variables as instruments for the dialect diversity and yield consistent evidence, proving the validity of our empirical findings. This study further discusses possible mediator variables between dialect diversity and rent-free farmland transfer. Corresponding mediation analysis indicates three types of mediation effects. First, the dialect diversity significantly increases the village farmland abandoned rate, and a higher village farmland abandoned rate urges rural households to choose rent-free farmland transfers rather than have their farmland left uncultivated. Second, dialect diversity also has negative impacts on the development of the local rural farmland market. A more active and developed farmland market reduces the probability of rent-free farmland transfers as it provides a better transaction platform for rural households to find suitable transaction partners. Rural farmland marketization gradually substitutes traditional farmland transactions. Third, dialect diversity is also closely related to transaction costs, which significantly promotes households' choices of more flexible farmland transfer contracts, such as oral contracts, especially during China's industrialization and urbanization process.

This study yields robust evidence, indicating that rural households are more likely to choose rent-free farmland transfers in a more diverse dialect-spoken region. As a special type of farmland transaction, households who receive the farmland for free also spare more effort in maintaining the farmland quality and repay the trust from their transaction partners. In less-developed rural areas, rent-free farmland transfers perform as an interconnected contract arrangement, which helps with local social communications. Additionally, as a type of flexible farmland transaction, rent-free farmland transfers let rural households have enough adjustment space to deal with their contracted land, especially during China's current industrialization and urbanization process, and rural farmland is generally expected to appreciate.

### 6.2. Discussion of Findings

In most of the previous related literature, rent-free farmland transfer is taken as the residual of the non-marketization of rural farmland transfers and is often criticized [1]. Though a standard and legally valid contract helps to curb the speculative behaviors of both parties during transactions, it will lead to the loss of adjustment flexibility. Many people worry that a flexible contract brings potential risks like the other side's speculative behaviors that may hurt their interests. However, it is almost impossible to have a perfectly designed contract, and an incomplete contract leaves space for opportunistic behaviors, which can induce moral hazard and adverse selection [63]. In rural communities, an incomplete contract is usually a mixture of agreement, relational contracts, and empty terms. The choices of each transaction party not only focus on the current timing of reaching an agreement but also manage to avoid renegotiation during the contract execution period, which saves unnecessary transaction costs of dealing with future uncertainties. Therefore, transactions using incomplete or flexible contracts, such as rent-free farmland transfer, are rational choices for both parties. The reciprocity mechanism and relationship network are built upon mutual trust and personal reputations in rural society [64], which ensures the stability of contracts by lowering the transaction costs [65,66] before, during, and after

farmland transfers. This study takes the rent-free farmland transfer as a special farmland transfer based on a more rent-flexible and operation-flexible contract compared to other regular farmland transfers.

Farmland transfer contracts have multidimensional factors, including rent, tenancy duration, lease status, buyer type, etc. Rent-free farmland transfer is a special type of informal contract, reflecting the farmers' rational choices and contractual trade-offs in the rural land market. According to the survey of 4156 rural households in Jiangxi and Guangdong Provinces, the rate of disputes within the subsample of choosing rent-free farmland transfers is only 2.16%, while this rate is nearly doubled within the subsample not choosing rent-free farmland transfers. Further, 73.15% of rent-free farmland transfers do not have a fixed term, which maintains contract flexibility for most rural households. Moreover, oral contracts of farmland transfer are also very common in rural land markets, at nearly 90% in 2005, and approximately half in 2018 during the gradual development of the rural land market, based on a nationwide survey [53].

China has a complex farmland tenancy system, as well as dialects, because of its vast territory and diverse customs and culture[6]. Farmland transfer is not only a transaction of economic factors but also a typical phenomenon of cultural interactions. Similarly, the institutional arrangements and implementations of agricultural land property rights should also take market theories and cultural traditions into account and form positive interactions. The great French philosopher Montesquieu pointed out that it is almost impossible to change cultural traditions simply through legal enforcement. For a long time, the informal system or social tradition has been an important supplement to the legal system [67]. The modern transformation of China's rural areas from a traditional society as well as the marketization of rural farmland elements still have a long way to go, which indicates that there is still plenty of room for the traditional resource allocation mechanism based on culture and conventions to improve [68].

The Chinese government places a strong emphasis on promoting the marketization and regulations of rural land transfer and on protecting the farmers' rights. However, as far as the marketization of farmland transfer is mentioned, policy documents from the Ministry of Agriculture and Rural Affairs or local governments focused on quantifiable indicators like formal contract signing, etc.; for example, 'they need to sign the written transfer contract based on consultation and agreement when the farmers' transfer cultivated land management rights[7]. This study shows that marketization is a gradual process, and culture and customs still have a significant role during farmland transfer.

### 6.3. Discussion of Limitations

There are some shortcomings in terms of theory and methods, because cultural research and informal institutions are very difficult fields, feeling like "blind men and the elephants" during the writing process. Future research might be expanded or deepened in certain directions, as below. Firstly, the characterization of regional culture should be a scientific definition and description, gradually from dialect diversity to a comprehensive index of regional culture. Secondly, we hope to collect nationwide panel data in future research. This study only focuses on two-year survey data from Guangdong and Jiangxi provinces due to limited funding. Thirdly, it is a very important aspect to explore culture and traditional social capital for market construction, while appropriate cultivation could maintain soil fertility in the long term.

**Author Contributions:** Conceptualization, S.L. and B.L.; methodology, S.L.; software, S.L. and Y.J.; validation, S.L. and B.L.; formal analysis, S.L.; investigation, B.L. and S.L.; resources, B.L.; data curation, B.L.; writing—original draft preparation, S.L., B.L. and Y.J.; writing—review and editing, S.L., B.L., Y.J. and X.Z.; visualization, S.L. and X.Z.; supervision, B.L.; project administration, B.L.; funding acquisition, B.L. All authors have read and agreed to the published version of the manuscript.

**Funding:** This research was funded by the National Social Science Foundation of China (Major Program), grant number: 23&ZD112.

**Institutional Review Board Statement:** This research is approved by Academic Ethics Committee of College of Economics and management, SCAU. Approval code: 20150128-1.

**Data Availability Statement:** The data presented in this study are available on request from the corresponding author, who will go through a legal application process and sign an agreement with the manager.

**Acknowledgments:** The authors extend great gratitude to the anonymous reviewers, academic editor, and managing editor for their helpful reviews and critical comments.

**Conflicts of Interest:** The authors declare no conflicts of interest.

## Appendix A

**Table A1.** Results of dialect diversity on rent-free farmland transfers based on two-stage least squares method (IV: Language Distance; Local Opera; War/Armed Confrontations).

| | Language Distance | | Local Opera | | War/Armed Confrontations | |
|---|---|---|---|---|---|---|
| **2nd Stage** | **Household (1)** | **Village (2)** | **Household (3)** | **Village (4)** | **Household (5)** | **Village (6)** |
| Dialect Diversity II | 0.0679 | 0.3259 | 0.2716 | 0.2013 *** | 0.3544 *** | 0.6592 *** |
| | (0.6221) | (0.2016) | (0.1727) | (0.0700) | (0.1241) | (0.0639) |
| Constant | 0.7039 | 0.5573 ** | 0.4596 | 0.8503 *** | 0.3586 | 0.2084 ** |
| | (0.7852) | (0.2406) | (0.2864) | (0.1145) | (0.2468) | (0.1148) |
| Control Variables | Yes | Yes | Yes | Yes | Yes | Yes |
| Wald Chi-square | 133.17 *** | 841.55 *** | 130.31 *** | 701.30 *** | 134.82 | 839.08 *** |
| Log pseudolikelihood/$R^2$ | −726.77401 | −1743.3694 | −754.14548 | −1892.7951 | −725.41666 | −1975.7959 |
| Wald/LM Test | 0.01 | 2.61 | 2.47 | 8.26 *** | 8.16 *** | 106.49 *** |
| *p*-value | (0.9131) | (0.1061) | (0.1158) | (0.004) | (0.0043) | (0.000) |
| Weak Instrument Test | 0.01 * | 2.61 | 2.48 | 8.17 *** | 8.24 *** | 120.20 ** |
| *p*-value | (0.9133) | (0.1059) | (0.1156) | (0.0043) | (0.0041) | (0.0000) |
| DHW Test | - | 0.6418 | - | 0.5140 | - | 87.906 *** |
| *p*-value | - | (0.4231) | - | (0.4735) | - | (0.000) |
| N | 1067 | 3103 | 1072 | 3133 | 1072 | 3133 |
| **1st Stage** | | | | | | |
| Instrument | 11.9827 *** | 17.2021 *** | 0.0551 *** | 0.0563 *** | 0.0264 *** | 0.0242 *** |
| | (4.1188) | (2.5671) | (0.1025) | (0.0027) | (0.0016) | (0.0010) |
| Constant | 0.8859 *** | 0.6152 *** | 0.0850 *** | 0.8482 *** | 0.9051 *** | 0.8703 *** |
| | (0.1578) | (0.0978) | (0.1025) | (0.0614) | (0.0979) | (0.0588) |
| Control Variables | Yes | Yes | Yes | Yes | Yes | Yes |
| $R^2$ | 0.3426 | 0.3312 | 0.4042 | 0.4054 | 0.4699 | 0.4280 |
| F-statistic | 36.51 *** | 101.90 *** | 47.77 *** | 141.67 *** | 62.41 *** | 155.48 *** |
| N | 1067 | 3103 | 1072 | 3133 | 1072 | 3133 |

Notes: Huber–White robust standard errors, in parentheses. *** $p < 0.01$, ** $p < 0.05$, * $p < 0.1$.

**Table A2.** Variables and descriptive statistics about instrumental variables and mechanism variables.

| Variable | Definition | Measurements | Obs | Mean | Std. Dev. |
|---|---|---|---|---|---|
| Instrumental variables | | | | | |
| IV1 | Degree of land relief of surveyed cities | Index | 4177 | 0.348 | 0.16 |
| IV2 | Annual average sunshine hours of cities | Hour | 4177 | 1718.571 | 116.196 |
| IV3 | Precipitation of cities | Millimeter | 4177 | 16,282.749 | 1203.661 |
| IV4 | Number of secondary tributaries in surveyed counties | Number | 4177 | 0.472 | 0.655 |
| IV5 | Historical path of dialects | - | 4177 | 4.509 | 1.938 |
| IV6 | 1/Language distance from Mandarin | - | 4132 | 0.022 | 0.003 |
| IV7 | Number of opera in surveyed cities | Number | 4172 | 3.341 | 1.602 |
| IV8 | Wars in surveyed regions | Number | 4177 | 10.706 | 4.589 |

**Table A2.** *Cont.*

| Variable | Definition | Measurements | Obs | Mean | Std. Dev. |
|---|---|---|---|---|---|
| Mechanism variables | | | | | |
| M1 | Village Farmland Abandoned Rate<br>% household farmland abandoned rate in each village | % | 4177 | 0.17 | 0.242 |
| M2 | Rural Farmland Market<br>Development Level<br>% transfer out or in farmland rate in each town | % | 4172 | 0.416 | 0.493 |
| M3 | Flexible Farmland<br>Transfer Contracts<br>0/1; equals 1 if transfer out with informal agreement, otherwise it equals 0 | 0/1 | 1726 | 0.721 | .0448 |
| Alternative dependent variables | | | | | |
| Rent-free in subsample of farmland net transfer-out | 0/1; equals 1 if farmers chose to rent out their farmland who only transfer-out farmland, otherwise, it equals 0 | 0/1 | 1077 | 0.301 | 0.459 |
| Net Farmland Transfer out | 0/1; equals 1 if farmers only transfer-out farmland at the first stage, otherwise it equals 0.<br>0.426 | 0/1 | 4177 | 0.239 | 0.426 |
| Village net rent-free Farmland Transfer out (%) | # of households only transfer out farmland for free / # of total household farmland net transfers out in a village | % | 3138 | 0.936 | 0.173 |
| Village Farmland net Transfer-out (%) | 0/1; equals 1 if Village has rent out farmland at the first stage, otherwise, it equals 0 | 0/1 | 4177 | 0.751 | 0.432 |
| Willingness to transfer out farmland rent-free | 0/1; equals 1 if the willingness of farmers to transfer out farmland for free, otherwise, it equals 0 | 0/1 | 1069 | 0.163 | 0.369 |
| Farmland Transfer-out (%) | 0/1; equals 1 if farmers chose to rent their farmland at the first stage, otherwise it equals 0. | 0/1 | 4177 | 0.256 | 0.436 |
| Willingness to transfer out farmland rent-free at village level | # Willingness of households only transfer out farmland for free/# of total household farmland net transfers out in a village | % | 3138 | 0.202 | 0.281 |

## Notes

[1] In 2014, the General Office of the State Council issued opinions on guiding the healthy development of the rural land use rights transaction market. In July 2016, the Ministry of Agriculture issued the regulations on the operation of the rural land management rights transfer market.

[2] Details can be referred to Policy and Reform Department of Ministry of agriculture and rural affairs. 2021 China Rural Policy and Reform Statistical Annual Report. China Agriculture Press: Beijing, China, 2022.

[3] People speak the same sub-class dialect in each dialect fragment.

[4] Details of the primary dependent variable, the key independent variables, and the control variables are displayed in Table 1. Details of the instrumental variables and the moderating variables are in Appendix Table A2.

[5] Estimation results related to Dialect Diversity I are also qualitatively consistent.

[6] Compiled by the Statistics Bureau of the Accounting Office of the Republic of China Government, 1946.

[7] Details can be referred to https://www.moa.gov.cn/govpublic/zcggs/202102/t20210203_6361060.htm (accessed on 26 February 2021).

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
