# Peer review of "The Impact of Dialect Diversity on Rent-Free Farmland Transfers: Evidence from Chinese Rural Household Surveys"

_land, doi:10.3390/land13020251_

Round 1
Reviewer 1 Report
Comments and Suggestions for Authors
This is an interesting study. Based on first-hand survey data and from the perspective of local culture, the author analyzes why there is a large number of free transfers in China's agricultural land transfer market, which has certain enlightening significance for the formulation and improvement of relevant policies of land transfer. In general, the research design is clear and reasonable, the method is used properly, and the conclusions are basically in line with common sense. It is suggested to publish after minor revision. Some suggestions are for reference:
(1) The introduction part needs to be moderately rewritten, especially to expand the review of the relevant research on the free transfer of cultivated land. At present, the author's entry into the relationship between local culture (dialect diversity) and the free transfer of cultivated land is a little too blunt and unconvincing. The premise of strengthening the persuasive force is to systematically describe the driving mechanism of the free transfer of cultivated land concerned by the existing research. Has anyone paid any attention to the influence of local culture on the transfer of cultivated land, and if so, to what extent? Only by giving these information clearly can the reader clearly grasp whether the author's marginal contribution is valid.
(2) This study lacks some necessary theoretical mechanism analysis and rigorous research hypothesis, so it is suggested that the author make some systematic expansion in the second part.
(3) Due to the lack of in-depth discussion in this study, it is suggested that the author add a discussion section, make an in-depth comparative analysis of the core results of the study and similar studies, and analyze the reasons for the differences. At the same time, the shortcomings of this study will also do some necessary expansion.
(4) The resolution of Figure 1 is not high, at the same time, there is no picture name in the picture, and now there is no specific information, it does not seem to matter.
Reviewer 2 Report
Comments and Suggestions for Authors
The article "The Impacts of the Dialect Diversity on Rent-free Farmland Transfers: Evidence from Chinese Rural Household Surveys” addresses an interesting topic, knowledge of which is very necessary for a better understanding of the factors that help or hinder land transfer processes in China. The writing style is clear and therefore relatively easy to read. In addition, the methodology is novel, correctly proposed and applied, and at the same time appropriate to the subject under study. For these reasons, this is a work that deserves to be published in a journal such as Land, in whose profile it fits well. Before doing so, however, the authors should make some improvements to the manuscript in order to correct certain weaknesses in the structural organization of the reasonings presented throughout the paper. Regarding these improvements, I make the following recommendations:
1) Between lines 49 and 61 (I am copying and pasting only the beginning and end of this paragraph) it says: "In China, each dialect-spoken (........) significance to unraveling the underlying mechanisms of China’s rural land market and its cultural basis.". This paragraph is a bit rambling and hard to understand if you want to take it literally, although the idea of what is meant seems clear, but you at the end you do not are sure to have understood it. For this reason, I recommend that the authors try to rewrite these lines in a less complicated and baroque style.
2) In addition, the introductory section needs to be rewritten in general, and care should be taken to make it clear what is to be studied. As currently written, this section does not make clear the relationship between dialectal diversity and rent-free land transfers. In principle, it would seem that such diversity would hinder transfers between those with different dialects and tend to facilitate them between those with the same dialect, i.e. between locals. This is indeed confirmed in later pages of the manuscript, but it should be made clear from the outset.
3) The section "2.1. Language and Regional Dialect” should be expanded a little further by trying to explain the relationship between Confucianism, local-regional dialects and local-regional identities. Writing a few additional paragraphs in this regard would allow readers who are not familiar with the Chinese reality to better understand the object of research of this article from its own introduction.
4) In one part of the manuscript, I leave it to the authors to decide which part, it should also be explained whether this continuing increase in land transfers translates into an increase in the cultivated area of plots.
5) I have suggested the above point because, in short, the manuscript seems to imply that it is intended to favor modernization and industrialization. This suggests that this concentration of plots is taking place as a result of the land transfer I mentioned earlier. A concentration that is somehow hindered by linguistic dialectal diversity.
6) Between lines 300 and 306 (I am copying and pasting only the beginning and end of this paragraph) it says: "Language communications of different species are also ecologically adaptive and impacted by climate factors, such as ambient humidity (…..) this paper applies annual local average sunshine hours and precipitation in our survey areas from 2000 to 2019 as instrumental variables of dialect diversity." Regarding this paragraph, I have to say that I find this kind of “environmental interpretation” of language dialectics characteristics very shocking and curious. On what and on whom do the authors rely to make such surprising and, in my opinion, unscientific statements? I lack sources and a better argumentation of what is said here.
7) Between lines 413 and 423 (I copy and paste only the beginning and the end of this paragraph) it says: "As China started its economic reform since 1978, an increasing number of rural labors persistently moved to urban areas for better-remunerated jobs and higher living standard (……,) As mentioned earlier, a higher regional dialect diversity increases difficulties of communication and cooperation, which can also affect local transactions and economic development". In relation to these lines, I would like to say to the authors that, in my opinion, this is where something becomes clear that should have been well and clearly explained in the Introduction from the beginning of the work.
8) The content of the article is not well organized and sequenced. To try to remedy this, I could put all the theoretical aspects of the manuscript at the beginning in the Introduction and/or the first sections. I would then focus only on the empirical research done.
9) The article is a bit long, so that as you read on, you find repetitions of ideas that have been said before. The authors should try to correct this by shortening the text as much as possible.
10) The current version of the manuscript lacks a short section outlining the Limitations of the manuscript. All research has limitations, and the authors should make it clear that they are aware of them. This section could also indicate future research along the lines developed in this paper.
11) Also missing is another short section dedicated to some recommendations for public policy makers responsible for the issues discussed in the article.
I hope that the above suggestions, which I have written with a basically constructive purpose, will serve as a guide for the authors to carry out a series of reforms in their manuscript, which will undoubtedly contribute to improving the quality, intelligibility and dissemination capacity of a work which, because of the subject it deals with and the way it deals with it, deserves to be disseminated as widely as possible throughout the world.
Reviewer 3 Report
Comments and Suggestions for Authors
The manuscript submitted for review addresses an important, interesting and current research problem corresponding to Land's profile.
Property rights, transaction costs, and non-contractual transfer of land are issues that are important and worthy of scientific consideration. This problem is also current in Poland and I encourage the authors to read the results of research on this topic conducted by Polish scientists.
The manuscript, as the authors rightly stated, contributes to previous literature in empirically studying the potential impacts from cultural factors, the dialect diversity, on rent-free farmland transfers. This paper applies modern econometric methods including the Heckman selection model and instrumental approach to solving potential sample selection issues and omitted variablebias in traditional OLS estimations. Based on a combined two-year rural household survey dataset, this paper provides micro-level evidence regarding rural households' farmland transfers as well as possible determining factors, which stands from a different perspective from previous regional or macro-level studies.
The peer-reviewed manuscript yields evidence indicating that rural households are more likely to choose rent-free farmland transfers in a more diverse dialect-spoken region. As a special type of farmland transactions, households who receive the farmland for free also spare more effort in maintaining the farmland quality and repay the trust from their transaction partners.
The selection of the research sample, its size and the method of selection are noteworthy. The authors are aware of the limitations of the study as written. The selection of the sample determined the further research procedure, which I evaluate positively. The selection of variables for the model also raises no objections.
The reviewer had some doubts regarding the discussion included in the manuscript. There is a need to analyze the research results of other scientists and compare them with the authors' research results (to the extent possible) or to emphasize the novelty of the research conducted and the limited possibilities of conducting a broader discussion.
The summary in its form does not reflect the essence of the scientific considerations presented in the manuscript. It is proposed to specify in detail, including an introduction to the research problem discussed in the manuscript, to formulate the purpose of the research, to briefly characterize the research methods and techniques used and the main research results. In the final chapter of the manuscript, it is worth referring to the research goal and its implementation.
Reviewer 4 Report
Comments and Suggestions for Authors
The comment is attached

Reviewer 5 Report
Comments and Suggestions for Authors
1. The abstract lacks basic methodological elements, i.e. the methods used, sources, research time and the exact spatial scope.
2. The introduction lacks a clear indication of the research problem and the validity of the research conducted, also in relation to the literature. The content requires a different order, e.g. the provinces are given first (line 46-47) and then it is repeated that they have been selected (89-90). Why were two provinces selected for research? From the point of view of the needs of an international reader, there is no short description of the research area and its characteristics.
3. The description of data sources in the methodological part mentions towns (lines 201-203). It also requires explanation (in the text of the study) why surveys from towns were used when writing about rural areas.
4. The weakness of the study is the lack of discussion of the results.
5. The lack of indication of the study's limitations is also disappointing.
